# Triangulation candidates for Bayesian optimization

**Robert B. Gramacy**
Department of Statistics
Virginia Tech
Blacksburg, VA 24060, USA
`rbg@vt.edu`

**Annie Sauer**
Department of Statistics
Virginia Tech
Blacksburg, VA 24060, USA
`anniees@vt.edu`

**Nathan Wycoff**
The McCourt School's Massive Data Institute
Georgetown University
Washington DC, 20057, USA
`nathan.wycoff@georgetown.edu`

## Abstract

Bayesian optimization involves "inner optimization" over a new-data acquisition criterion which is non-convex/highly multi-modal, may be non-differentiable, or may otherwise thwart local numerical optimizers. In such cases it is common to replace continuous search with a discrete one over random candidates. Here we propose using candidates based on a Delaunay triangulation of the existing input design. We detail the construction of these "tricands" and demonstrate empirically how they outperform *both* numerically optimized acquisitions and random candidate-based alternatives, and are well-suited for hybrid schemes, on benchmark synthetic and real simulation experiments.

## 1 Introduction

We address the continuous unconstrained optimization problem

$$x^\star = \underset{x \in \mathcal{B}}{\operatorname{argmin}} f(x) \tag{1}$$

where the bounding box $\mathcal{B}$ is a hyperrectangle, often taken as $[0,1]^d$ in coded inputs. The objective $f : \mathcal{B} \to \mathbb{R}$ is a *blackbox* function, meaning that we can only learn about its behavior through expensive, often simulation-based, evaluation. Such problems are most challenging when $f$ is highly non-convex, and thus contains multiple local minima. A tacit goal of a solver is to minimize the number of times that $f$ is evaluated in search of a global solution.

The earliest papers on Bayesian optimization (BO) adapted statistical modeling and design principles to tackle this optimization problem [Močkus, 1975, Jones et al., 1998]. Applications on physics-based simulators $f$ are provided by Pourmohamad and Lee [2021]; scenarios in machine learning are reviewed by Garnett [2022]. BO is common in studies of engagement and user experiences in online platforms [e.g., Letham and Bakshy, 2019], hyperparameter estimation for deep learning [e.g. Turner et al., 2021, Feurer et al., 2018] and materials design [e.g. Zhang et al., 2020b], to name a few.

To illustrate BO and introduce our contributions, consider $f(x) = \sin(x)$ and $\mathcal{B} = [-1, 2\pi + 1]$ evaluated at six equally-spaced inputs $x$. Next fit a surrogate $\hat{f}_n$, to data $(X_n, Y_n)$, where $y_i = f(x_i)$, for $i = 1, \ldots, n = 6$. We privilege Gaussian process (GP) based-surrogates [e.g., Gramacy, 2020],

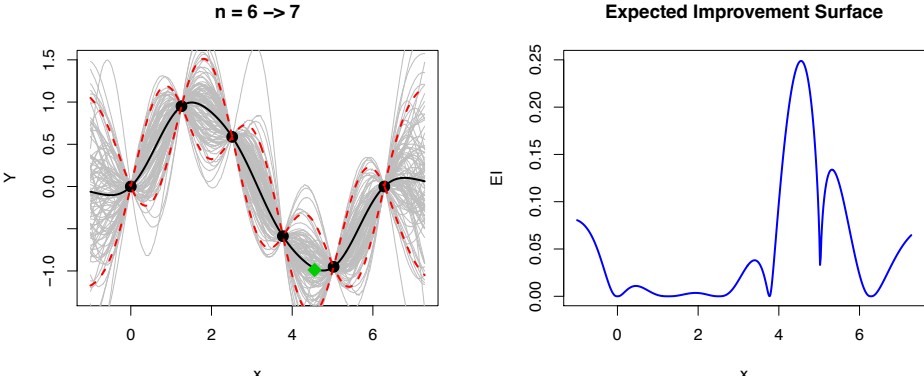

Figure 1: Predictive surface (left) via means (solid black), 90% intervals (dashed-red), and sample paths (gray); EI (right) and resulting acquisition (green diamond).

but our methodology is agnostic to that choice so long as the predictive equations from $\hat{f}_n$ have similar features – e.g., non-linear predictive mean $\mu_n(x)$ and higher predictive uncertainty/standard deviation $\sigma_n(x)$ away from the training data sites.

The left panel of Figure 1 shows a fitted $\hat{f}_6$ via $\mu_6$ as solid black line with error bars $\mu_6 \pm 2\sigma_6$ in dashed-red. Each of the one-hundred gray lines is a sample from the predictive distribution. This distribution interpolates the training data (black dots), a GP hallmark, yet our contributions are not limited to noise free settings. (We entertain a noisy $f$ in Section 3.2.) The current best run $f_n^{\min} = \min\{y_1, \ldots, y_n\}$ is near $x = 5$, but there is considerable predictive uncertainty about other parts of the input space $\mathcal{B}$, relative to $f_n^{\min}$. Locations in $[-1, 0] \cup [4, 6] \cup [2\pi, 2\pi + 1]$ are promising.

BO *acquisition criteria* serve to operationalize this notion. We shall focus on two and note that several others are variations on similar themes. One is *Thompson sampling* [TS; Thompson, 1933], and involves drawing from the predictive distribution $Y(x) \sim \hat{f}_n \mid X_n, Y_n$ and choosing $x_{n+1} = \text{argmin}_{x \in \mathcal{B}} Y(x)$. TS amounts to randomly selecting one of those gray lines and optimizing it in lieu of the expensive $f$, so the criterion is inherently stochastic. The other is *expected improvement* [EI; Jones et al., 1998]. Define *improvement* as $I(x) = \min\{0, f_n^{\min} - Y(x)\}$ and take its expectation with respect to $Y(x)$. If $Y(x) \sim \hat{f}_n$ is Gaussian, as with GPs, then this has closed form:

$$\text{EI}(x) = \mathbb{E}\{I(X)\} = \int I(x)\, dY(x) = (f_n^{\min} - \mu_n(x))\, \Phi(z_n(x)) + \sigma_n(x)\, \phi(z_n(x)), \quad (2)$$

where $z_n(x) = (f_n^{\min} - \mu_n(x))/\sigma_n(x)$ and $\Phi/\phi$ are the Gaussian cdf/pdf. For non-Gaussian $\hat{f}_n$, one can always resort to Monte Carlo (MC) integration instead. The right panel of Figure 1 provides EI for $\hat{f}_6$. Contrary to TS, the EI acquisition $x_{n+1} = \text{argmax}_{x \in \mathcal{B}} \text{EI}(x)$ can be deterministic, if properly solved. The maximum EI acquisition $x_7$ is shown as a green diamond on the left panel.

Observe that TS and EI involve "inner-optimizations" over a criterion, a task that may be even more challenging than the original problem (1). Each gray line is highly multi-modal, as is the EI surface. Both have about as many local optima as there are training data points $n$, whereas $f$ has only two local minima in $\mathcal{B}$. Speedy evaluation of $\mu_n(x)$ and $\sigma_n(x)$ relative to $f(x)$ saves us, but only partly. We still desire good $x_{n+1}$ without exhaustive search or cumbersome subroutines.

Two strategies are common, sometimes separately, sometimes in tandem as a hybrid. The simplest option is to distribute $N$ candidate points $\mathcal{X}_N$ throughout the input space, evaluate the criterion on $\mathcal{X}_N$, and thereby replace a continuous search with a discrete one. In low input dimension a dense grid of candidates can be used effectively. In higher dimension one can populate $\mathcal{X}_N$ with a space-filling design like a random Latin hypercube sample [LHS; Mckay et al., 1979] to manage the computational expense of evaluating the criterion exhaustively. A higher-powered approach is to locally apply a smooth, convex optimization library such as L-BFGS-B [Byrd et al., 2003]. Derivatives may be approximated by finite-differencing or autograd [Paszke et al., 2017], or may have a simple closed form depending on the criterion and nature of $\hat{f}_n$. The former requires more computer work and some consideration of numerical stability; the latter more researcher/programmer effort when possible. (MC-based $\hat{f}_n$ or EI challenges both approaches.) Pure candidate-based search is most common with

TS because of its stochastic nature. Gradient-based continuous search is popular in simple GP/EI setups, but a multi-start scheme is essential to avoid inferior local solutions. This is where the hybrids come in: candidates seeding local solvers.

In this paper we contend that both strategies, random candidates and multi-start local optimization, can be replaced by (or hybridized with) more thoughtfully chosen $\mathcal{X}_N$. In the 1d setting of Figure 1 we could place candidates at the midway points between each of the existing $n$ inputs and the boundary $\mathcal{B}$, implementing a kind of multi-pronged bisection search [Burden and Faires, 1985, Section 2.1] and resulting in $N = n + 1 = 7$ candidates. The best of those $\mathcal{X}_N$ by either criterion may not give a precise solution to the inner optimization, but it would be an effective one because EI and many of the random gray lines indicate a solution close to one of those midway points. Such locations might be much better than ones identified by a limited candidate or numerical local search.

This 1d example is overly simplistic. Going forward, we shall explicitly target two and higher dimensions. In Section 2 we scale-up the midway candidate idea, also suggested by Scott et al. [2011], to what we call "tricands", based on Delaunay triangulation and the convex hull of $X_n$. We explore tricands' features and limitations and suggest remedies with the BO application in mind. In Section 3 we demonstrate that tricands outperform both random LHS candidates and a multi-start gradient-based numerical inner optimization in the conventional setting where surrogate GP predictive equations are available in closed form. In Section 4 we consider two nonstationary surrogates requiring Markov chain Monte Carlo (MCMC), for which closed-form acquisition criteria are not readily available. Candidates are essential in this setting, and our tricands are better than random space-filling ones. Our discussion in Section 5 emphasizes tricands' "plug-n-play" nature – they can be inserted into any candidate-based scheme – and suggests potential for extension.

## 2 Delaunay triangulation candidates

Many criteria for BO resemble Eq. (2), balancing exploitation ($\mu_n(x)$ below $f_n^{\min}$) with exploration (large $\sigma_n(x)$). Most surrogates inflate predictive uncertainty ($\sigma_n(x)$) away from training data locations $X_n$. In the case of GPs, this is what produces the "sausage shaped" predictive intervals shown as red-dashed lines in Figure 1. Our main insight is that careful allocation of candidates *between* existing training data locations, where $\sigma_n(x)$ is high, allows for BO acquisitions that do not necessitate cumbersome numerical optimization of posterior predictive quantities but still balance exploitation and exploration. A hard statistical optimization can be replaced with an easier geometric one.

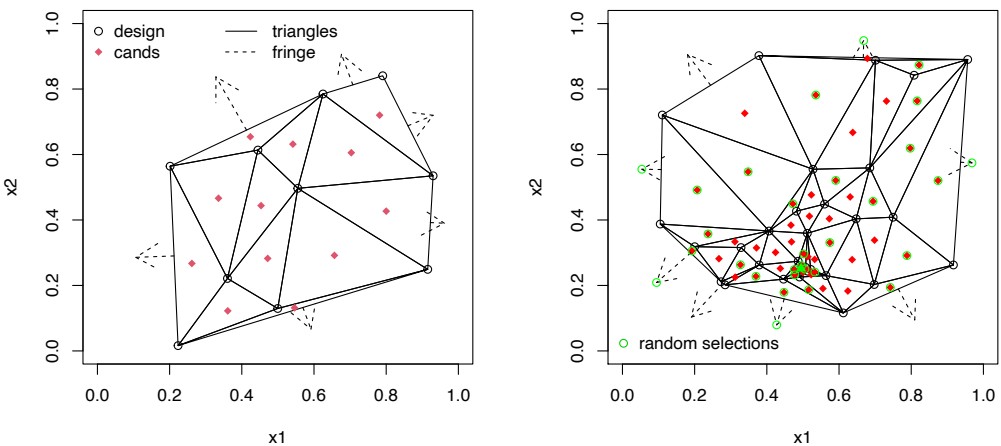

Figure 2: Interior and fringe candidates (both), and randomly sub-sampled candidates (right).

The idea is sketched graphically in Figure 2, whose details will emerge over the next three subsections. The existing design $X_n$ is shown as open circles, and the selected candidates are closed red dots *and* the tips of dashed arrows. We call these interior (Section 2.1) and fringe (Section 2.2) candidates, respectively. Both are calculated based on Delaunay triangles and their convex hull, outlined by solid black lines in the figure, and explained next. We provide this illustration in 2d to ease visualization; however our Section 3–4 benchmarks include higher dimensional input spaces.

## 2.1 Interior candidates

A *Delaunay triangulation* of $X_n$ is an angle-maximizing set of $d$ line-segments connecting geographically nearby points, such that no point lies inside the circumcircle of those points. Such triangulations subdivide the interior of the *convex hull* of $X_n$, which is the the smallest (convex) set that contains all points. In 2d, lines depicting those subdivisions form triangles. In higher dimension they form tetrahedra, however one often abuses the nomenclature and still refers to triangles. For more details, see, e.g., Lee and Schachter [1980]. The solid lines of Figure 2 indicate a triangulation for random $X_n$ (left) and an $X_n$ derived after BO iterations (right), described in Section 3.

There are fast algorithms for calculating Delaunay triangulations. In 2d, these have $\mathcal{O}(n \log n)$ runtimes. Higher dimensional analysis is complicated by the number of triangles, which depends on the geometry of $X_n$, a topic we shall return to shortly. For R we use the geometry package on CRAN [Habel et al., 2019]; for Python we use Delaunay in scipy.spatial [Virtanen et al., 2020]. Both are wrappers around the C library Qhull, implementing "quickhull" [Barber et al., 1996]. Bates and Pronzato [2001] first suggested Delaunay triangulation for BO. That early work only explored two input dimensions, possibly because they did not have convenient access to Qhull. They also did not entertain the BO-specific extensions we provide here, particularly in Sections 2.2–2.3.

Let the triangles be denoted by $T_j$, for $j = 1, \ldots, n_T$. Each $T_j$ is a $(d+1) \times d$ matrix when $X_n$ has $d$ columns. Create $n_T$ new candidates $\mathcal{X}_{n_T}$ where the $j^{\text{th}}$ candidate is located at the *barycenter* of $T_j$:

$$\tilde{x}_j = \bar{T}_j = \frac{1}{d+1} \sum_{i=1}^{d+1} T_j[i,] \quad \text{or} \quad \tilde{x}_{jk} = \frac{1}{d+1} \sum_{i=1}^{d+1} T_j[i,k], \quad \text{for } k = 1, \ldots, d.$$

The left expression is vectorized over the second, column dimension of $T_j$. The second is explicit about coordinates $\tilde{x}_j^\top = (\tilde{x}_{j1}, \ldots, \tilde{x}_{jd})$. Red dots in Figure 2 provide a visual. These locations will almost certainly not be the maximal points of $\sigma_n(x)$ in the vicinity of $T_j$, but they will be close because $\tilde{x}_j$ is within $T_j$ but far from its edges. We refer to these $\mathcal{X}_{n_T}$ as "interior" candidates.

In 2d, Euler's formula gives that $n_T = 2n - 2 - h(X_n)$ where $h(X_n)$ is the number of elements of $X_n$ on its convex hull. In Figure 2, $n = 10$ and $h(X_n) = 6$ so $n_T = 12$. When $d \geq 3$, the number of faces of the tetrahedra can grow as $n^{\lceil d/2 \rceil}$ depending on the nature of $X_n$. Figure 3 provides

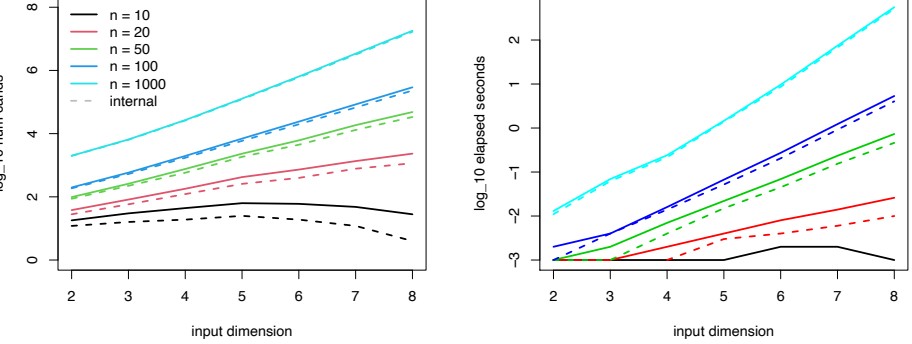

Figure 3: Tricands $N$ (left) over input dimension $d$ and design size $n$, and compute time (right) using a single core of an Intel i7-6900K CPU at 3.20GHz.

an empirical view of the number of candidates (left) and triangulation compute time (right), for varying $d$ and $n$ where $X_n$ is distributed uniformly at random in $\mathcal{B} = [0,1]^d$. The dashed lines in the left panel are $\log_{10} n_T$. Notice that when $n$ is small, $n_T$ decreases as $d$ increases. For fixed $d$, $n_T$ steadily increases with $n$. In our BO context this is a good thing, because so too does the modality of acquisition criterion. With nonparametric surrogates like GPs, the complexity of the response surface can increase with training data size $n$. The search effort for the next acquisition should be commensurate with this complexity, an innate characteristic of (interior) tricands. The right panel of Figure 2, described later in Section 2.3, shows how this works after many iterations of EI acquisition.

## 2.2 Fringe candidates

Having candidates $\mathcal{X}_{n_T}$ only in the interior of the convex hull could limit exploration if the complement of the volume of the hull and $\mathcal{B}$ is large. It could limit exploitation if the solution is on/near the boundary of $\mathcal{B}$. One remedy is to force $X_n$ to contain boundary points, such as the corners of the input space. This is not a bad approach in low dimension (e.g., in 2d there are only four corners), but could be prohibitive in higher dimension. Later in Section 3 we consider a $d = 8$ example, having 256 corners in $\mathcal{B} = [0, 1]^8$, which would nearly blow our entire budget of runs.

We instead prefer the "fringe" candidates pointed to by the dashed arrows in Figure 2. There is one of these for each facet (edge in 2d) of the convex hull, extending perpendicularly from middle of the facet half way to the boundary of $\mathcal{B} = [0, 1]^d$. The Qhull library furnishes both ingredients: $d \times d$ facets $F_j$ and normal $d$-vectors $\vec{v}_j$ for $j = 1, \ldots, n_F$. Using these quantities, let $\bar{F}_j = \frac{1}{d} \sum_{i=1}^{d} F_j[j,]$ denote the coordinates of the middle of facet $j$. Then, the $j^{\text{th}}$ fringe candidate in $\mathcal{X}_{n_F}$ is

$$\vec{x}_j = \bar{F}_j + \tfrac{1}{2}\alpha_j \vec{v}_j \quad \text{where} \quad \alpha_j = \min \left\{ \mathbb{I}_{\{\vec{v}_j > 0\}} - \bar{F}_j \vec{v}_j \right\}.$$

Above, "min" is picking out the nearest boundary to $\bar{F}_j$. Division by two mirrors the bisection search analogy for interior candidates, but this could be a tuning parameter. Non-unit rectangular $\mathcal{B}$ is doable, but requires a more convoluted formula. More general $\mathcal{B}$ may present challenges.

Fringe and interior candidates may be combined: $\mathcal{X}_N = [\mathcal{X}_{n_T}; \mathcal{X}_{n_F}]$, stacked row-wise to form an $N \times d$ matrix with $N = n_T + n_F$. The number of fringe candidates $n_F$ is generally small compared to $n_T$. This is shown empirically in the left panel of Figure 3, where $N$ is indicated by the solid line, and $n_T$ by the dashed one. As $n$ increases, the gap between $N$ and $n_T$, indicating $n_F$, narrows.

## 2.3 Targeted sub-sampling

Having $N$ grow exponentially in $n$ may not suit all applications. Entertaining $N \approx 10{,}000$ candidates when $n = 100$ and $d = 6$, referring to Figure 3, is cumbersome and potentially overkill. One way to limit $N$ is random sub-sampling: simply calculate the full $\mathcal{X}_N$ based on $X_n$ and downsample, uniformly at random, a subset of size $N_{\text{sub}} < N$. Since our triangulation strategy was designed to focus on exploration (finding locally high $\sigma_n(x)$) via midway candidates, we find it advantageous to guarantee retaining some of those candidates which are promising for exploitation (low $\mu_n(x)$), still without any explicit numerical optimization – only geometry.

One of the rows of $X_n$ corresponds to $f_n^{\min}$, the best input found so far: $x_n^{\min} = x_i$ s.t. $i = \operatorname{argmin}_{i=1,\ldots,n} y_i$ (or $\mu_n(x_i)$ in the noisy case). Let $\mathcal{T}_n^{\min} = \{T_j : x_n^{\min} \in T_j, j = 1, \ldots, n_T\}$ denote the set of triangles containing $x_n^{\min}$, and similarly let $\mathcal{X}_n^{\min} = \{\tilde{x}_j \in \mathcal{X}_N : T_j \in \mathcal{T}_n^{\min}\}$ denote the candidates associated with those triangles. Those are, in a geometric sense, adjacent to $x_n^{\min}$. A fringe candidate may also be considered adjacent in an analogous way, however we place them in the complement $\mathcal{X}_N \setminus \mathcal{X}_n^{\min}$. Now, rather than sub-sample uniformly from the full set $\mathcal{X}_N$, we partition sampling from points adjacent to $x_n^{\min}$, i.e., from $\mathcal{X}_n^{\min}$, and from points farther afield in $\mathcal{X}_N \setminus \mathcal{X}_n^{\min}$. In so doing, we guarantee that our $N_{\text{sub}}$ candidates cover potential for exploitation and exploration, respectively. We prefer a 10:90 split, with up to 10% of $N_{\text{sub}}$ coming from candidates adjacent to $x_n^{\min}$, fewer if $|\mathcal{X}_n^{\min}| < N_{\text{sub}}$, and likewise 90% from its complement.

The right panel of Figure 2 provides an illustration after $n = 30$ runs optimizing the Goldstein–Price function with EI under random initialization (details in Section 3.1). When $n = 30$ we have $N \approx 60$, with the precise value depending on $h(X_n)$. Here we consider $N_{\text{sub}} = 30$. Observe how randomly sub-sampled candidates $\mathcal{X}_{N_{\text{sub}}}$, circling the original red candidates $\mathcal{X}_N$ in green, concentrate near the global minimum $(0.5, 0.25)$ because $x_n^{\min}$ is in the vicinity. Other sub-sampled candidates are spread out more widely. This figure illustrates how acquisitions, and thus candidates, gravitate toward promising regions for exploitation without neglecting areas of potential exploration. A ridge of local minima may be found in the banana-shaped region traced out by a concentration of $X_n$ and $\mathcal{X}_N$.

## 2.4 Implementation and software

Our implementation, provided for Python and R in our git repository,[1] is relatively tidy. For example, `tricands.R` therein contains just 71 lines of code, eleven of which are to support optional

---

[1] http://bitbucket.org/gramacylab/tricands

visualizations in 2d such as those in Figure 2. The heavy lifting is done by `Qhull`. When performing Delaunay triangulations, we provide option `"Q12"` to work around numerical instabilities that sometimes arise. When calculating convex hulls, option `"n"` returns normal vectors used in calculating fringe candidates (Section 2.2).

Defaults yield both fringe and interior candidates and fix a maximum candidate size of $\texttt{max} = N_{\text{sub}} = 100d$, but these are user-adjustable. In experiments coming shortly, we deliberately limit $N_{\text{sub}}$ even further so that a fairer comparison can be made to other continuous search and candidate-based methods. When $N \leq N_{\text{sub}}$, all interior and fringe candidates are returned. The user may supplement these with additional $N_{\text{sub}} - N$ random candidates if desired. If $N_{\text{sub}} < N$, the execution flow looks for a variable `best`, providing the index of $x_n^{\min}$ in $X_n$, making sure about 10% of tricands come from adjacent triangles. When `best=NULL`, tricands are sub-sampled at random.

## 3 Classical GP benchmarking

In this first of two sections on benchmarking, we focus on BO via traditional GP surrogates. We follow a homework problem in Section 7.4 of Gramacy [2020], which piggy-backs off of GP, EI, and TS demonstrations earlier in the chapter. Software and other particulars are relegated to Appendix A.1. All of our examples are fully reproducible using the code provided in our git repository. We consider three methods for solving EI acquisitions: a continuous search of the criterion via L-BFGS-B with 5-multi-starts, LHS candidates, and tricands. For TS acquisitions, we similarly employ both LHS candidates and tricands. As non-BO (not surrogate-based) comparators, serving primarily as benchmarks, we entertain "raw" L-BFGS-B and Nelder–Mead [Nelder and Mead, 1965].

Two examples are showcased here, with a third example relegated to Appendix A.2. Our synthetic $f$s, including those in Section 4, are described in more detail on the pages of the Virtual Library for Simulation Experiments [VLSE; Surjanovic and Bingham, 2013]. In all of our experiments we code inputs to $[0, 1]^d$ and summarize results for 100 random restarts where each surrogate is initialized with the same (unique to each random restart) starting design of size $n_0 = 12$, except in Section 3.2 where we use $n_0 = 60$. This design is taken uniformly at random following the advice of Zhang et al. [2021] who caution that small space-filling initial designs can spark pathological behavior in BO. We track "best observed value" (BOV) $f_n^{\min}$ as a measure of progress which is summarized by median over $n = 1, \ldots, n_{\text{end}}$ and by boxplots for particular $n$ along the way.

### 3.1 Goldstein–Price

The Goldstein–Price function is a popular low-dimensional (2d) benchmark for BO [e.g., Picheny et al., 2012]. A total of $n_{\text{end}} = 50$ acquisitions are entertained, and all candidate-based methods (tricands and LHS) are limited to fifty candidates. This means $\texttt{max} = N_{\text{sub}} = 50$ for tricands, with fewer candidates when $N < N_{\text{sub}}$. The top row of Figure 4 summarizes results in three views. In the top-left panel, median progress in BOV is shown over increasing budgets of evaluations ($n$) as if each subsequent acquisition were the last. In the top-middle and right panels, boxplots capture the distribution of BOV at $n = 30$ and $n = 50$, respectively. In the top-left panel, methods based on a multi-start numerical local search use solid lines; those based on tricands are dashed; those based on LHS candidates are dotted. (Nelder–Mead is an exception, being red-dashed.) In the boxplots, tricands use slightly heavier ink so that they stand out. Text printed at the top of the top-right panel indicates the average number of times the criterion, EI or TS, was evaluated in solving the inner optimization sub-problem(s), cumulative over all acquisitions. In the case of a numerically optimized EI (labeled "EI"), these happen within iterations of L-BFGS-B search.

Beginning with medians over $n$ in the top-left panel, observe that the dashed lines (those based on tricands) are uniformly superior to all other comparators. EI with tricands is slightly better than TS with tricands. Only at the very end of the run does the raw Nelder–Mead alternative look competitive. At the 30th evaluation (top-middle) the boxplots indicate that there are some MC repetitions where BOV based on tricands are underperforming. However, more than half of the time these are the best two (i.e., both EI and TS with tricands) of all. TS with tricands is the winner in terms of best-case performance, whereas the EI version has slightly better worst-case behavior. Neither local optimizer (red) is competitive. Finally, in the top-right panel we see that the story is similar, except that now EI with tricands edges out its TS analog. Although median performance for Nelder–Mead is competitive, more than half of the time its BOV at $n_{\text{end}} = 50$ is among the two worst in the experiment.

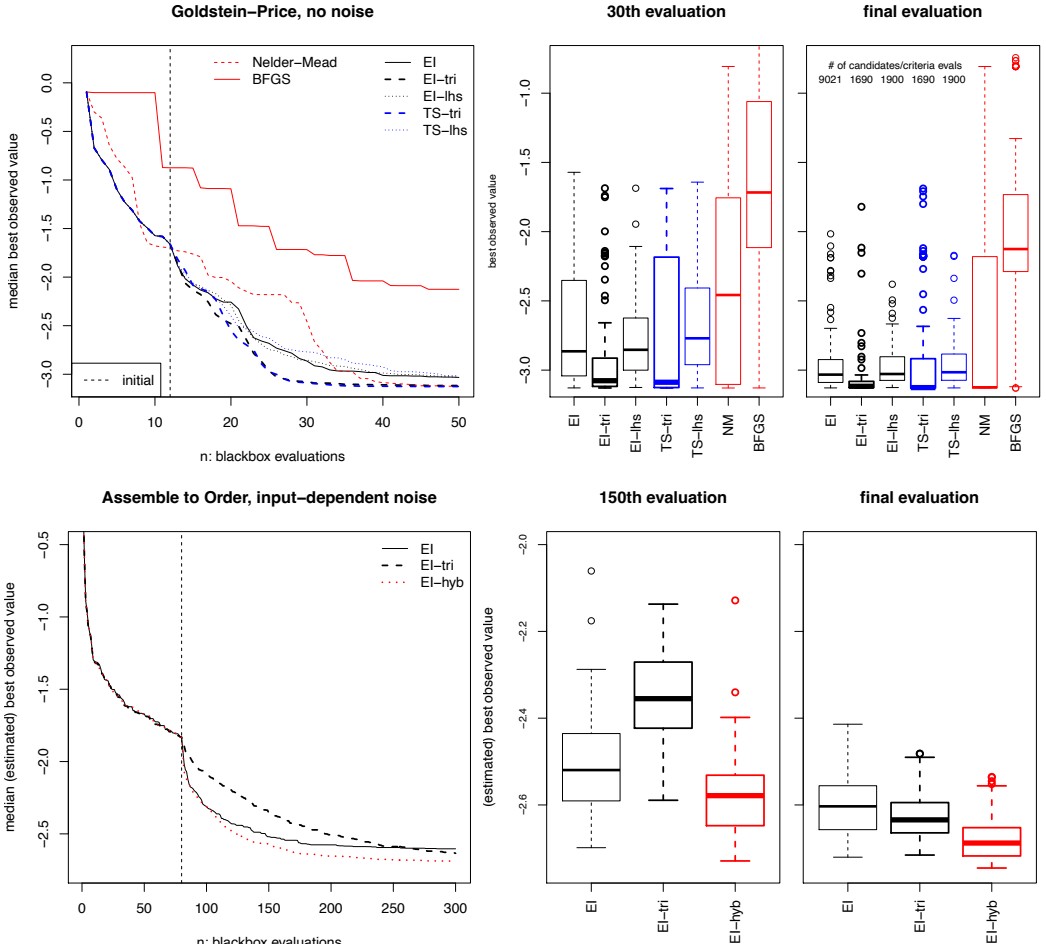

Figure 4: Goldstein–Price (top) and ATO (bottom) BOV over 100 MC trials: median (left panel) and distribution at intermediate (middle) and final (right) acquisitions. Suffixes "-tri" or "-lhs" denote a candidate search. Absence of a suffix (e.g., "EI" alone) indicates multi-start L-BFGS-B.

Perhaps the most striking result from the figure is that tricands-based EI ("EI-tri") outperforms its multi-start numerical analog ("EI") despite a factor of five fewer evaluations of the EI criterion (about 9000 compared to 1700). A more aggressive, gradient-based search misutilizes computational resources. Finding a precise solution – ultimately furnishing a maximal local value in an inferior domain of attraction – may come at the expense of finding an accurate approximation near a global solution. The same is true, but to a lesser extent, when comparing TS variations (5% reduction).

See Appendix A.2 for nearly identical results with the higher dimensional Hartmann 6 function.

### 3.2 Assemble to order (ATO)

The assemble-to-order (ATO) simulator [Hong and Nelson, 2006] deploys a queuing system virtualizing a manufacturer receiving orders for a variety of widgets, each with different demand for component parts. Target inventory levels for eight parts comprise the controllable inputs ($d = 8$). The output is profit, a scalar distilling inventory and order fulfillment costs under random orders for parts, fulfillment and inventory replenishment times. ATO is coded in Matlab and exhibits input-dependent noise. Although it is reasonably fast (seconds per evaluation), relatively large $n$ is required to separate signal from noise ($n_0 = 80$ and $n_{end} = 300$). Cubic computational bottlnecks for GP inference are in play here, despite using a thrifty heteroskedastic GP [Binois et al., 2018]. More details, including software, are provided in Appendix A.1. Since our Matlab licenses prevented us from running parallel instances, we had to limit our experiment somewhat to obtain results in a reasonable amount

of time. Therefore we only entertained three acquisition alternatives: random multi-start numerically optimized EI, tricands with the default $N_{\text{sub}} = 100d = 800$, and a hybrid using the best tricands candidate to initialize a local numerically optimized EI ("EI-hyb").

The bottom panels of Figure 4 track BOV for these three alternatives in a view similar to the panels above. Note that the original ATO problem involves maximization, so we have negated the output and also scaled it so that its units were more similar to our other benchmark problems. Since the output is random, the BOVs are estimates. We used the surrogate fit at $n_{\text{end}}$ to assign in-sample predictions via $\mu_{n_{\text{end}}}(x_i)$. Otherwise, the story is similar to our earlier synthetic results. Tricands' progress is initially slower than numerically optimized EI, but eventually gives lower BOV values. The difference at $n_{\text{end}}$ is slight, but statistically significant. A paired Wilcox test with alternative hypothesis that BOV for "EI-tri" is below "EI" has a $p$-value of 0.0037. The hybrid is even better, beating pure tricands and pure numerical optimization 85 and 88 out of 100 times, respectively.

## 4 Sampling-based surrogates

Tricands are most valuable when the inner-optimization problem cannot be solved by library-based numerical methods, even locally. This happens when the surrogate predictive surface is discontinuous and/or when inference requires MCMC. Our examples here involve response surfaces exhibiting nonstationarity, meaning that the input–output dynamics evolve over the input space. This demands a more elaborate surrogate. We consider two. A treed Gaussian process [TGP; Gramacy and Lee, 2008] uses axis-aligned partitioning with GPs. Such divide-and-conquer excels when dynamics change abruptly across individual inputs creating distinct regimes. MCMC can average over the location of probable partition boundaries. We use the R package `tgp` on CRAN [Gramacy, 2007], following Section 4 of Gramacy and Taddy [2010] for candidate-based EI acquisition.

Deep Gaussian processes [DGPs; Damianou and Lawrence, 2013] warp inputs to accommodate a more subtle evolution of dynamics in the input space compared to the abrupt regime changes of TGP. Although variational inference is popular for DGPs [Salimbeni and Deisenroth, 2017], Sauer et al. [2020] argue that in active learning contexts, such as BO, full posterior integration leads to better uncertainty quantification, and thus better acquisitions. Here we use the R package `deepgp` on CRAN [Sauer, 2021] which supports EI evaluation on candidates.

Unfortunately, neither `tgp` nor `deepgp` facilitate TS acquisition for BO. However, the `tgp` package furnishes a maximum *a posteriori* sample which can be used as the basis of a local search [Gramacy and Taddy, 2010, Section 4]. Here we show that tricands provides better candidates for both pure candidate EI search and this hybrid scheme.

### 4.1 Abrupt changes

The Gramacy & Lee (G&L) function benefits from a model supporting hard breaks even though its dynamics evolve smoothly. The 2d input domain (coded to $[0, 1]^2$) is mostly flat except in one quadrant where a local maximum is twinned with a (global) minimum. The domain of attraction of that global minimum covers only about 10% of the input space. It is easily missed by random initial designs and candidate sets. TGP is able to isolate the interesting quadrant with just two axis aligned partitions, thus recognizing that sampling effort should be concentrated there.

The top row of Figure 5 shows results in the same three views as earlier. The number of candidates is limited to twenty, otherwise the setup is unchanged. First ignore the red lines and boxplots (those labeled "hyb-") and focus on the black (TGP) and green (DGP). Notice that in both cases the tricands-based comparators, dashed and/or bolded, outperform their LHS-based analogues. In terms of medians over $n$ (top-left panel), that dominance is uniform. In terms of boxplots, the disparity is stark with the exception of a few outlying DGP-based BOV values with tricands. TGP seems to edge out DGP (with tricands) which we attribute to the abrupt change between the interesting quadrant and the rest of the input space. Now focus on the red "hyb-" results. These are the ones where TGP candidate-based search is finished with a gradient-based EI search on the maximum *a posteriori* model. Again, tricands wins. To supplement the visuals, we report that the tricands hybrid BOV value was lowest in 97/100 reps (top-right).

A higher dimensional example using the Michaelwicz function is provided in Appendix A.3.

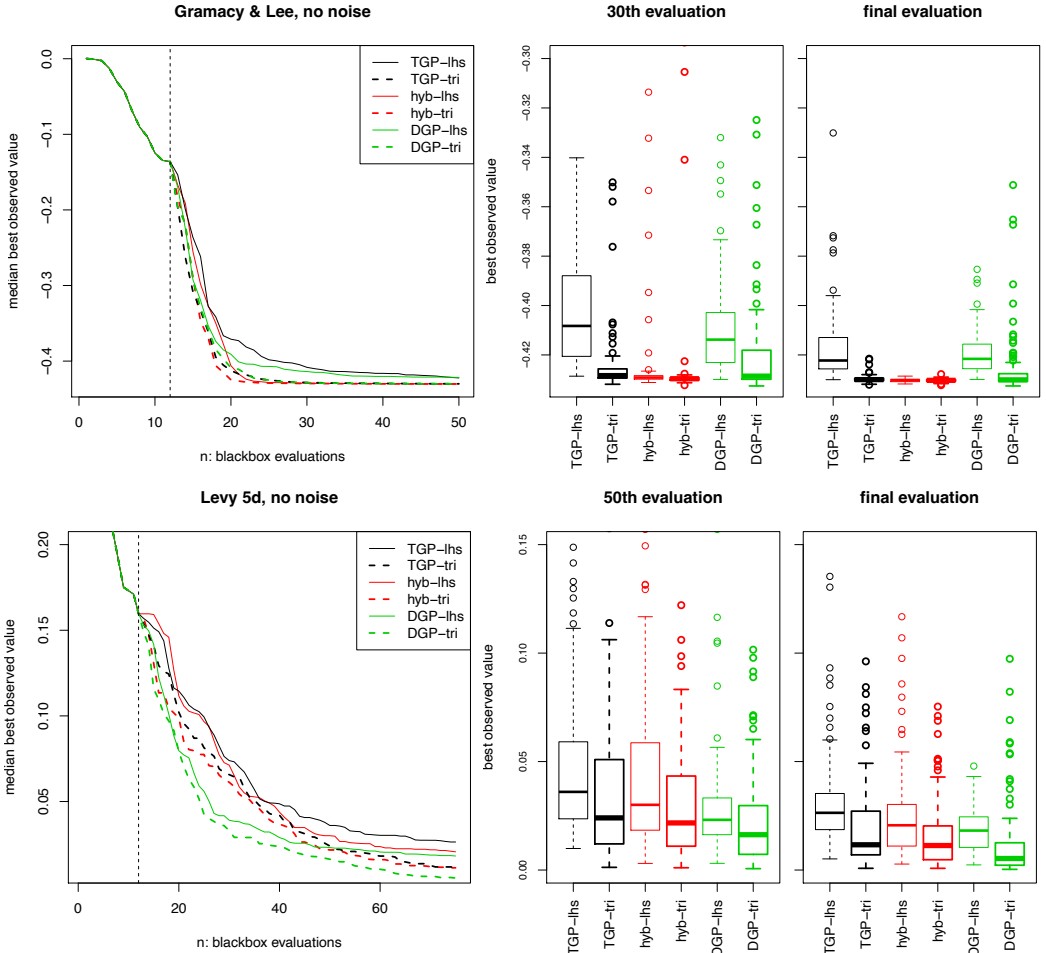

Figure 5: Gramacy & Lee (top) and 5d Levy (bottom) with MCMC-based comparators.

## 4.2 Smooth changes

The Levy function is defined in arbitrary dimension. It is ruffled with many peaks and valleys. Here we consider $d = 5$ for variety; see the bottom row of Figure 5. Commensurate with our other examples, we consider two hundred candidates and $n_{\text{end}} = 75$.

Both candidate DGP variations outperform their TGP counterparts. Slowly/smoothly varying dynamics favor warping inputs as opposed to hard partitioning. This is evident in both median and full distribution (boxplot) views. In the median view (bottom-left), notice that the dashed lines are uniformly under the solid ones of the same color. Tricands are providing better median performance over all $n$. Although the preeminence of tricands is apparent visually, we report that it provided better BOV in 81/100 MC restarts among the best (DGP) comparators. When tricands are involved, the hybrid candidate/numerical option is not discernibly better than the pure candidate alternative.

## 5 Discussion

We offer a novel take on space-filling candidates for acquisition in BO. The idea is to fill the spaces in-between previous acquisitions. This is motivated by an analogy to bisection search, but also by the nature of GP predictive surfaces which often serve as surrogates in BO. GP surrogates have organically inflated uncertainty between training data sites, which makes those spots attractive for BO acquisition. This notion is extended to the space between the convex hull of existing training data and the boundary $\mathcal{B}$ of the study region. In an array of benchmark exercises we have demonstrated that these tricands lead to superior performance in BO compared to both higher-powered gradient-based

acquisition schemes and simpler space-filling candidates. Tricands' main attraction is that they mimic the behavior of higher-powered searches with the implementation simplicity of candidates. That simplicity means that tricands may be deployed where the higher-powered alternatives cannot, such as when the surrogate is not continuous or requires MCMC.

Additional discussion on high input dimension and other extensions is provided in Appendix B.

## Acknowledgments and Disclosure of Funding

We are grateful for funding from DOE LAB 17-1697 via subaward from Argonne National Laboratory for SciDAC/DOE Office of Science ASCR and High Energy Physics, and for comments from two teams of anonymous referees which greatly improved the paper.

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
