# A  Implementation and additional empirical results

Here we summarize implementation details and experimental results that were removed from the main body of the paper due to space constraints. All of the empirical work in our paper is fully reproducible. Code may be found in our git repository: http://bitbucket.org/gramacylab/tricands.

## A.1  Classical GP implementation details

The GP surrogate for the Goldstein–Price and Hartman 6 examples (Section 3 and Appendix A.2, respectively) used the `laGP` package for R [Gramacy, 2016, Gramacy and Sun, 2018] on CRAN with separable/automatic-relevance determination lengthscales in a Gaussian kernel using gradient-based MLE. We supplied our own EI function, following the prototype provided in Chapter 7 of Gramacy [2020], modified to keep track of the number of evaluations. Finite-differencing was used to approximate gradients. For Latin hypercube samples we used the `lhs` [Carnell, 2018] package for R. "Raw" comparators L-BFGS-B and Nelder–Mead [Nelder and Mead, 1965] local optimizers were facilitated via the `optim` function in R.

The heteroskedastic GP surrogate used for ATO (Section 3.2) was via `hetGP` [Binois and Gramacy, 2021] for R. The built-in EI capability in that package, which supports both numerical optimization (via analytic gradients) and candidates, unfortunately could not easily be modified to keep track of the number of evaluations. Simulations, which require Matlab, were run in R through the R.matlab interface [Bengtsson, 2018]. The input space for ATO is discrete, in $\{1, \ldots, 20\}^8$, which is different from the real-valued inputs of our other examples. The only modification we made to accommodate this nuance was to "snap" acquisitions to that grid, implemented in coded units. Sometimes this resulted in replications in the design. No additional consideration was given to replicates, despite findings that they are beneficial in similar contexts [Binois et al., 2019].

Details for the surrogates used in Section 4 are provided therein.

## A.2  Hartmann 6

As a second example of conventional GP-based BO, consider the six-dimensional Hartmann function. Again, see Picheny et al. [2012] for more on this benchmark. Our experimental setup is identical the Goldstein–Price experiment in Section 3 except here we use the default $N_{\text{sub}} = 100d = 600$. Figure 6 summarizes results. Even with this high $N_{\text{sub}}$, tricands still result in many fewer acquisition criteria evaluations than numerically optimized EI (right panel), which must search more aggressively for the local optimum in such high dimension.

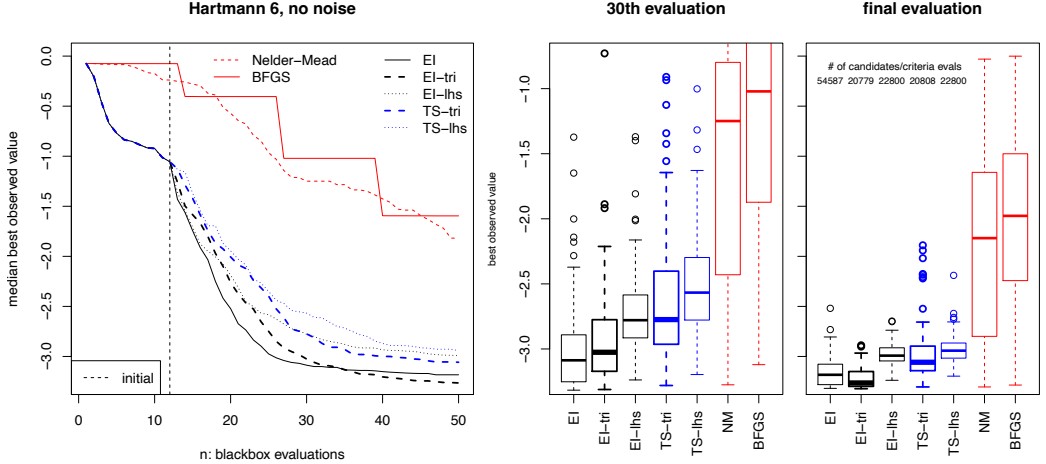

Figure 6: Analog of Figure 4 for the Hartmann 6 function.

The results are broadly similar to our earlier 2d Goldstein–Price example shown in Figure 4. Tricands-based EI yields equivalent, or better, BOV despite many fewer criteria evaluations. This is true, but to a lesser extent, with TS. It is notable that tricands came from behind in the case of EI. For the first

twenty or so acquisitions, numerically optimized EI bests its tricands analog. Geometric bias towards exploration may be less desirable in early acquisitions, but pays off by the end of the search.

## A.3  Michaelwicz

As a second example of abruptly changing regimes, continuing from Section 4.1, consider the Michaelwicz function. Like G&L, the surface has large flat areas, but it also has a continuum of ridges of local minima which intersect to create deeper valleys of local minima, and ultimately one global minimum where the deepest of those ridges intersect. A nice feature of the Michaelwicz function is that it is defined in arbitrary input dimension. Here we use it in 4d, which makes for a very difficult surface to model and optimize. To cope, we take the search out to $n_{\mathrm{end}} = 75$ and allow up to two hundred candidates per acquisition. Otherwise the setup is similar to earlier examples.

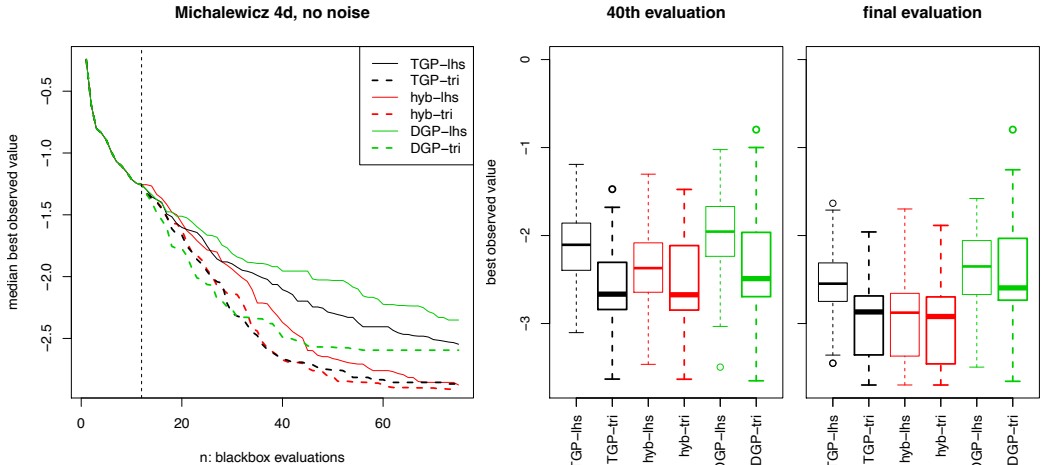

Figure 7: Michaelwicz function in 4d with MCMC-based comparators.

Figure 7 shows the results, which are largely similar to G&L, except revealing of these additional challenges. Tricands uniformly dominate LHS via median BOV. Although noise is high across all boxplots, reflecting variability in solution quality over random re-initialization, the patterns are clear in a pairwise analysis. For example, the red boxplots in the right panel look similar but, at the 75th evaluation, 75/100 tricands-based BOVs were below their LHS counterpart. As in the G&L example, TGP bests DGP. Abrupt regime changes are at odds with the DGP's smooth warping of inputs.

## B  Additional discussion

When reflecting on tricands' value, it is important to remember some stylized facts about BO. One is that the theory (under stringent regularity conditions) guarantees convergence to the global minimum "eventually," in the sense that with enough samples you'll explore everywhere [e.g., Bull, 2011]. That's not much help in practice, because exploring everywhere isn't practical. Another is that EI and TS are greedy; their scope is the next acquisition. Contemplating a remaining budget and entertaining optimal decisions through that lens can be caricatured as a herculean effort with marginal gains [Gonzalez et al., 2016, Frazier et al., 2008, Lam et al., 2016, Gramacy and Lee, 2011]. These are not bad ideas, but they haven't moved the needle on the *modus operandi* because they've not been incorporated into accessible libraries, and are too involved for bespoke implementation in practice.

Both theory and aggressively scoped acquisition can lose sight of the real goal. What's important in BO, and active learning in general, is creating a virtuous cycle between data acquisition and learning. Anyone who has tried knows that there is a fine line between vicious and virtuous when it comes to implementation details, despite the best of intentions and theoretical "guarantees". EI and TS are simple to implement because GP libraries are in abundance, code evaluating criteria is a few lines long, and there are many examples to cut-and-paste from. Barriers to application are low, but it's easy to get carried away to disappointment. This is what makes tricands attractive. It is motivated by simple principles: the solution is in-between your current runs, so look there. It is plug-n-play

wherever candidates are an option. This means they can be applied in situations where numerical differentiation is not available (e.g., with MC integrated surrogate). Evaluation of the acquisition criteria can be massively parallelized with candidates, say on a GPU, whereas local numerical solvers are inherently sequential.

That's not to say that tricands are a panacea. Some of our boxplots indicated that improvements over alternatives on best-observed-value (BOV) in 90/100 MC repetitions may have come at the expense of the worst case performance of the remaining 10%. This may have simply been bad luck. But if not, the deterministic nature of tricands could be to blame: not enough opportunity to be surprised. As mentioned in the main body of the paper, one can always augment tricands with random/space-filling candidates. This could be especially beneficial when $N$, the number of tricands for a given $n$, is lower than the desired budget of candidates. Entertaining tuning parameters for some of our hard-coded settings, like the distance between fringe candidates and the boundary (Section 2.2), could help. That distance could even be chosen at random. We could likewise randomly choose a location for interior candidates (Section 2.1), within each triangle, rather than taking the barycenter. When randomly downsampling (Section 2.3), we could guarantee a certain proportion of fringe candidates like we did for ones adjacent to $f_n^{\min}$.

Although the important subroutines of Delaunay triangulation and convex hulls are off-loaded to libraries, they can (at times) be computationally demanding. When $n$ is large and $d$ is modest, calculating $N$ locations in the thousands (see Figure 3) could be cumbersome. Of course, the whole goal of BO is to limit $n$. In our experiments with $n \leq 75$, all of our triangulation/hull calculations took fractions of a second. But with big $n$ they can take minutes (Figure 2, right panel), and that could be prohibitive. However, after each acquisition the number of new sites only increases by one ($n \rightarrow n + 1$), and thus affects only a small, local part of the triangulation/hull. The Qhull library does not support this, but there are incremental algorithms for triangle/hull augmentation which are very fast relative to starting from scratch. For more details, see Su and Drysdale [1997].

Such a strategy might be valuable in higher dimensional BO settings. Continuous optimization theory tells us that gradient-based methods have local convergence rates independent of dimension, which has made L-BFGS-B and similar optimizers the tool of choice in solving for acquisitions. This, together with the exponential growth of input space volume, might at first blush suggest that tricands' performance is limited to low/modest dimension. However, in practice, the highly nonconvex nature of the acquisition surface significantly cheapens the theoretical results associated with gradient-based optimization. Indeed, recent work has achieved state-of-the-art performance in high $d$ using candidate sets focused within a certain region of the input space [Eriksson et al., 2019, Wang et al., 2020, Daulton et al., 2021], though still built on traditional space-filling points such as LHS. It would be interesting to see whether replacing these space-filling points with tricands would be as beneficial in that setting as we have found it to be in modest $d$. Furthermore, a popular approach in scaling BO to high dimension is to reduce the input space by screening input variables or finding linear or nonlinear embedding spaces, rendering the problem a low dimensional one (see Binois and Wycoff [2021] for an overview). The aim of such an approach is in part to make solving the acquisition problem easier, and there's no reason to believe this wouldn't extend to tricands.

Our surrogates were GP-centric, extended to handle non-stationarity via treed partitioning and smooth (deep GP) input-warping. It would be interesting to explore the value of tricands paired with more unconventional surrogates based on trees, such as random forests [Breiman, 2001] or tree-structure Parzen estimators [Bergstra et al., 2011], where inner-optimization via gradient-based local search is a non-starter. We presented results with EI and TS-based acquisition criteria, and of course there are a litany of other heuristics. Our early experiments additionally included the upper-confidence bound [UCB; Srinivas et al., 2009] and probability of improvement (PI) criteria. The former, for most settings of the tuning parameter, mirrored our EI results whereas the latter was dominated by EI. To reduce clutter, we decided not to include them in our presentation here.

Our ATO example in Section 3.2 involved a stochastic simulator with input-dependent noise. Surrogate modeling and active learning for stochastic simulation is still very much on the frontier of the computer experiments landscape [Baker et al., 2020]. In such settings, the acquisition space should be extended to include the possibility of obtaining a replicate run, exactly duplicating one of the $n$ existing design elements [Binois et al., 2019]. Replication can be advantageous in separating signal from noise in generic active learning tasks, and specifically in the context of BO [Binois and Gramacy, 2021, Section 4.2]. Rather than entertaining a hybrid search between a continuum of novel

locations and a discrete set of replicate sites, tricands could be leveraged to make the entire set of candidates discrete, vastly simplifying the inner optimization search.

Batch acquisition, acquiring several new runs at once, is a common paradigm in some settings. Ideas with modified EI go back at least to Ginsbourger et al. [2007] with several following thereafter [e.g., Taddy et al., 2009, Chevalier and Ginsbourger, 2013]. One, more recent approach involves penalizing regions of the input space near earlier acquisitions in the batch [González et al., 2016]. This spirit could be ported to a tricands setting: simply rule out any candidates which are in triangles $T_j$ adjacent to those acquired earlier in the batch.

Finally, although we have emphasized BO, other active learning criteria could benefit from candidates well-spaced relative to the current design. Perhaps the most common is for active learning targeting reduced integrated mean-squared prediction error [IMSPE, e.g., Leatherman et al., 2017, Binois et al., 2019, Zhang et al., 2020a]. High input variance locations, such as between design sites, are natural candidates for reducing IMSPE. Another recently popular active learning topic in computer experiments is contour/level set finding [Ranjan et al., 2008, Bect et al., 2012, Chevalier et al., 2014, Marques et al., 2018, Azzimonti et al., 2020, Cole et al., 2021]. Many of the criteria suggested in these works involve predictive entropy from a GP surrogate, which is famously myopic; entropy (defined via a classification above and below a level set) tends to be higher near training data already near partition boundaries, leading to a clumping of acquisitions unless explicit measures are taken to spread out candidates, or to otherwise deter a numerical inner-optimizer. Tricands could offer a geometric spread of future acquisitions away from existing training data locations.