# OpenReview forum: "Triangulation candidates for Bayesian optimization"
_NeurIPS.cc/2022/Conference — NeurIPS 2022 Accept_

### Official Review · Reviewer_eznT · 2022-07-08

**Rating:** 8
**Confidence:** 5
**Soundness:** 4 excellent
**Presentation:** 4 excellent
**Contribution:** 4 excellent

**Summary:**

The paper introduces a method for optimizing the acquisition function during Bayesian optimization. The method is based on geometry, rather than numerical optimization or random search, the two most common techniques. It generalizes the well-known "midway candidate" idea that has been used as an acquisition optimization initializer by at least 2 software packages, and turns it into a complete strategy that works in full-dimensional Bayesian optimization problems. The results with the new method are very promising and outperform gradient optimization.

**Questions:**

The questions are those above: How do total wall times compare for the methods, and how does tricands perform without targeted subsampling?

**Limitations:**

Yes

**Strengths And Weaknesses:**

The paper is very well written and addresses an important problem. Writers of Bayesian optimization software have put a level of effort into this problem that is not fully reflected in the literature, and so it is easy to underestimate the impact of this paper. But it tackles a very important problem, and one that underlies all of the results and progress in the field. It thus has broad impact in the entire space of Bayesian optimization.

The method presented is exceptionally innovative. The general idea of the "midway candidate" is well-known and has been used as an intializer for gradient optimization, but previous applications of this idea have been limited and not extended naturally to higher dimensions, which is why it has only been successful as an _initializer_ for a subsequent, expensive gradient optimization. The paper uses concepts from geometry to turn this idea into a method that applies to the full range of Bayesian optimization problems, which will be especially useful for extended Bayesian optimization to surrogate models and acquisition functions in which gradients may be difficult to compute. The level of novelty in this paper is very high, for Bayesian optimization particularly.

The results are very impressive and show that we can get _better_ performance than gradient optimization without having to compute gradients. This is a very strong result that I expect will change the way people think about acquisition optimization.

There are three areas where I think the paper could improve:

* Evaluation is given in terms of number of candidates evaluated, which is a great thing to directly compare. But it would also be helpful to see a comparison in the overhead (for gradient optimization this means computing gradients / matrix operations, for tricands it means the triangulation). Seeing the total wall-time for each method would allow that.

* How important is the targeted subsampling? Seeing some results without targeted subsampling to see would be helpful, as well as some study of the 10% number.

Both of these are minor revisions.

---

> ### Author Response · Authors · 2022-08-01
> **Author Response**
>
> Thanks so much for your review.
>
> We agree that it could be helpful to provide timings alongside our progress metrics.  We note that for most of our examples, tricands takes fractions of a second to calculate whereas GP surrogate evaluation generally takes longer (orders of magnitude longer for TGP and Deep GPs), as does EI solving.  It could help to be more explicit about that.
>
> How much targeted subsampling helps is problem specific, and seems to help more in lower dimensions.  We agree it would be good to explore this more fully.

---

### Official Review · Reviewer_czZM · 2022-07-10

**Rating:** 5
**Confidence:** 4
**Soundness:** 2 fair
**Presentation:** 3 good
**Contribution:** 1 poor

**Summary:**

The authors consider the use of triangulations to generate space-filling candidates in Bayesian optimization. They consider a few low-dimensional synthetic problems and find that their triangulation-based approach as well as a hybrid approach marginally outperforms EI with numerical optimization.

**Questions:**

- How do you initialize optimization with multi-start L-BFGS-B and how many restarts do you do? People would generally use a large number of quasi-random candidates here, e.g., 2048, as computing the acquisition function values for simple acquisition functions such as EI is relatively cheap. Using an unreasonably small number of initial points may explain why EI flattens out in Figure 4 (bottom) as it could be that all starting points have acquisition function value zero.
- Packages like Spearmint and BoTorch provide the option to generate candidates as local perturbations around the best point found so far which often works well in practice. This is often important later on in the optimization run when exploitation becomes more important and finding candidates with non-zero acquisition function value becomes more challenging. Can you comment on why you didn't consider such an approach and how the performance compares to using triangulations?
- That numerical optimization of the acquisition function (which depends on the posterior) is cumbersome is a bit vague and unmotivated. The acquisition function surface is in many cases be easy to optimize (EI, UCB, etc.), especially in cases where the gradient can be computed analytically. Can you comment on what you find to be cumbersome here?
- The intuition that we want to select candidates where sigma(x) is high sounds reasonable in the low-dimensional setting as long as it is supplemented with candidates that also favor exploitation as those will be important later in the optimization run. As an alternative to using triangulations, do you think it would be possible to screen candidates based on the minimum distance to the training points as that would provide a cheaper but potentially still effective way to select good candidates?

**Limitations:**

Yes

**Strengths And Weaknesses:**

Strengths:
- This is generally a well-written paper that very clearly explains the background material and motivates the approach.
- The successes of multi-start L-BFGS-B is indeed sensitive to the choice of good starting points. This is particularly important when using acquisition function like EI where a large majority of the domain may have zero acquisition function value, particularly later on in the optimization run. The authors observe that the use of triangulations lead to a small performance improvement.

Weaknesses:
- The approach is limited to low-dimensional problems (8 is the largest considered in the paper) as the use of triangulations is well-known to scale poorly with the input space dimensionality.
- I don't think this paper brings much value to the BO community. In particular, the authors don't compare to alternative approaches for generating candidates, which makes it hard to judge the value of the approach.

---

> ### Author Response · Authors · 2022-08-01
> **Author Response**
>
> Thanks so much for your review.
>
> We wish to mention that we do indeed compare our approach to alternative candidate schemes in the form of a Latin Hypercube Sample (LHS), which our experiments indicate to be inferior.
>
> The purpose of tricands is not to outperform EI on vanilla GPs necessarily (though it does happen to), but rather to provide a thrifty approximation applicable to sophisticated, probabilistic surrogates, such as the deep GP.
>
> To address the referee’s four bulleted questions individually:
> 1. We conduct multi-start optimizations initialized with points from a random LHS with twice as many points as there are input dimensions.
> 2. Our approach of subsampling while favoring triangles which contain the current optimal point was designed to achieve a similar end.
> 3. While EI may have a closed form acquisition function, its nonconvexity and steep changes make it difficult to optimize in practice, especially for complicated surrogates.
> 4. Though we need a way of generating candidates in the first place, we agree it may be interesting to screen them based on distance to the existing design points.

---

> > ### Comment · Reviewer_czZM · 2022-08-09
> > **Thank you for your comments**
> >
> > I have read the rebuttal and the feedback from the other reviewers. I'm satisfied with most of the responses except for (1) as I don't think anyone would use so few initialization points given that, e.g., EI is well-known to be numerically zero in large parts of the domain. For example, Ax (which relies on BoTorch) uses 20 restarts and 1000 initial points by default which is more likely to yield good starting points and allow you to optimize EI successfully using a multi-start framework. Given how you currently initialize the acquisition function optimization it isn't at all that surprising you beat EI + a vanilla GP, but I wonder how these results would change given a larger number of initial random (spray) points. I suggest that the authors incorporate these suggestions in the CR version of the paper, assuming it gets accepted. Anyway, this is a nice idea and a well-written paper so I have increased my score from 4 to 5.

---

### Official Review · Reviewer_xvts · 2022-07-11

**Rating:** 7
**Confidence:** 4
**Soundness:** 4 excellent
**Presentation:** 4 excellent
**Contribution:** 4 excellent

**Summary:**

The paper aims to improve the optimization of an acquisition function in Bayesian optimization (BayesOpt).
The motivation stems from the observation that a typical acquisition function trades off between exploration and exploitation and therefore is maximized in a region between or far away from observations, but never close to them.
The authors propose a scheme to generate candidates for the search to optimize the acquisition function that relies on Delaunay triangulation, generating two sets of candidates: interior points that lie between observations and fringe points that lie outside the convex hull of the observed set.
To improve efficiency, the algorithm also employs a non-uniform sampling strategy from these two sets to assemble a final candidate set to evaluate the acquisition function with.
Experiments across benchmark objective functions show that the proposed algorithm leads to better optimization performance for two common acquisition functions: expected improvement and Thompson sampling.
Finally, the paper investigates situations in which the predictive model produces predictions that make the acquisition function unamenable to gradient-based optimization, such as treed or deep Gaussian processes, showing that their algorithm consistently improves performance across the different models.

**Questions:**

The motivation of the algorithm, looking between the observations as well as regions far away, feels to me similar to a space-filling objective where we'd like to, in addition to the observations already made, cover the rest of the space.
Have the authors considered a baseline of this kind where it generates points that together with the points in the training set, infills the space and uses generated points as candidates for optimizing the acquisition function?
I expect this strategy would bias regions far away from observed points and might over-explore, but it may still pose as a spiritual competitor to highlight the benefits of the proposed algorithm and its subsampling scheme.

On the topic of subsampling, a potential enhancement could be to avoid regions with high predictive mean (in the minimization problem in our case) when sampling the interior points, since those points most likely do not maximize the acquisition function anyway.

**Limitations:**

Yes, the paper mentions its limitations in the appendix.

**Strengths And Weaknesses:**

This is a good submission.
The exposition flows well, and the question of how to optimize the acquisition function in BayesOpt is important yet often ignored.
The proposed algorithm is natural and well-motivated.
It is clear from their experiment results that their policy consistently performs well across many tasks compared to other baselines.
I also enjoyed reading the addition discussions in the appendix, which make connections to other problems and areas.

I don't have any major complaints about this submission.
One thing that the paper could benefit from is experiments with higher-dimensional ($d > 20$) objective functions.
I would be interested to see the performance and running time of the algorithm, as well as whether there's any change in behavior, when there are a large number of dimensions and triangulation is presumably more difficult.
I talk more about a potential baseline that the authors may consider comparing the algorithm against in the Questions section.

Overall, the submission addresses an important problem and the proposed solution seems to work well on the inspected settings, and I lean towards acceptance.

---

> ### Author Response · Authors · 2022-08-01
> **Author Response**
>
> Many thanks.
>
> We have begun to tinker with higher input dimensions, exploring tricks similar to those in TuRBO, etc.  About infill candidates, this is exactly what tricands is doing, ensuring infill in all triangles.  One can certainly infill in otherways, such as via maximin, but that can be computationally intensive and avoids areas where the solution is because points are clustered there already.  We’ve seen poor performance from conventional infilling in preliminary studies, but we agree such a benchmark would provide for a more comprehensive empirical comparison. We also agree that the targeted subsampling could be enhanced by discarding large mean values.

---

### Official Review · Reviewer_NN2f · 2022-07-15

**Rating:** 6
**Confidence:** 1
**Soundness:** 2 fair
**Presentation:** 3 good
**Contribution:** 2 fair

**Summary:**

This paper proposes a new space-filling acquisition function for Bayesian optimization based on Delaunay triangulation of the initial data.

**Questions:**

-

**Limitations:**

-

**Strengths And Weaknesses:**

The idea of using Delaunay triangulation to get new observation locations is intuitively appealing, and the paper clearly explains their method. However the paper, as it stands, is barely adequate. No theoretical analysis justifies their choices, the experimental analysis is unconvincing, and the literature survey is meager.

---

> ### Author Response · Authors · 2022-08-01
> **Author response**
>
> Thank you. While we do not consider a three page bibliography to be meager, we welcome any suggestions for its expansion. Additionally, we would welcome any suggestions for extensions to our experimental analysis. Finally, we would welcome any suggestions for theoretical properties to investigate.

---

> > ### Comment · Reviewer_NN2f · 2022-08-08
> > **Changed score**
> >
> > After going over the relevant literature and the paper again, I am increasing my score from 3 to 6, and decreasing my confidence score from 2 to 1. I missed the importance of the contributions of this paper on my first read.

---

### Meta-Review · Area_Chair_DoWo · 2022-08-25

**Recommendation:** Accept
**Confidence:** Certain

**Metareview:**

Overall, I think this paper makes a pretty interesting contribution. I think that one of the fundamental problems of Bayesian optimization that is often swept aside (at least in the literature if not in software) is the fact that--with many acquisition functions--it doesn't really solve global optimization as a problem, but merely shifts that problem around to a much less expensive one (e.g., optimizing the cheap acquisition function rather than the cheap objective function). Despite this, extremely simple procedures like restarted gradient descent methods and discretization are pretty common place, and very little progress has been made or even attempted on this problem in recent years. The authors approach here seems pretty reasonable based on a relatively agreeable intuition.

With that said, I do think it would be very useful for the authors in the camera ready version to address Reviewer czZM's final comments in their last remark. I agree that 2*d initializations for multi-start optimization is extremely small for most problems, especially since modern software can support optimizing from each initialization in batch mode parallel pretty efficiently. Given that multi-start optimization is arguably the predominant method used at least in full software implementations, it's probably worth including a comparison at a variety of initialization budgets, up to *much* larger ones than this.

**Award:**

No

---

### Decision · Program_Chairs · 2022-09-14

Accept